# Development of Molecular Tools to Identify the Avocado (*Persea americana*) *West-Indian* Horticultural Race and Its Hybrids

**DOI:** 10.3390/ijms262311510

**Published:** 2025-11-27

**Authors:** Mario González Carracedo, Samuel Bello Alonso, Anselmo Ramos Luis, Ainhoa Escuela Escobar, David Jiménez Arias, José Antonio Pérez Pérez

**Affiliations:** 1Departamento de Bioquímica, Microbiología, Biología Celular y Genética, Área de Genética, Universidad de La Laguna, 38200 La Laguna, Canary Islands, Spain; mgonzalc@ull.edu.es (M.G.C.); samuelbelloalonso@gmail.com (S.B.A.); aescuela@ull.edu.es (A.E.E.); 2Instituto Universitario de Enfermedades Tropicales y Salud Pública de Canarias, Universidad de La Laguna, 38200 La Laguna, Canary Islands, Spain; 3Instituto Canario de Investigaciones Agrarias, Ctra. de El Boquerón s/n, Valle Guerra, 38270 La Laguna, Canary Islands, Spain; aramos@icia.es; 4Agroquímica, ICIA, Unit Associated with CSIC by IPNA and EEZ, 38270 La Laguna, Canary Islands, Spain

**Keywords:** avocado (*Persea americana*), horticultural races, *West-Indian*, salinity tolerance, molecular markers, retrotransposon-based insertion polymorphism (RBIP)

## Abstract

Avocado (*Persea americana* Mill.) is cultivated in a wide range of environments, from tropical and semitropical to subtropical regions. Its fruit, of high nutritional value, is increasingly demanded worldwide. Spain is the main European producer, but avocado cultivation in certain areas, such as the Canary Islands, requires the genetic identification of *West-Indian* rootstocks because they often show tolerance to low-quality water and soil salinization. In the present study, eight novel Retrotransposon-Based Insertion Polymorphism assays, derived from previously characterized inter-Primer Binding Site markers, have been developed and evaluated by multiplex PCR across 58 *P. americana* cultivars. The results showed 100% specificity and sensitivity in detecting the *West-Indian* genomic component, both in pure and hybrid avocado cultivars. This cost-effective and fast molecular tool provides a valuable resource for characterization and selection programs of avocado cultivars genetically related to the *West-Indian* horticultural race.

## 1. Introduction

Avocado (*Persea americana*) is considered one of the most agronomically important species worldwide. It is predicted to be the world’s second-best-selling tropical fruit by 2030 [1]. Since 2019, European avocado imports have grown by approximately 10–20% annually, thus highlighting the importance of developing new agricultural production strategies within Europe. Spain is currently the leading European avocado producer, with the crop having been established in the Canary Islands, Málaga, and the Granada coast, and continually expanding into new areas such as the Valencian Community, Cádiz, Huelva, and the Cantabrian coast, replacing less profitable crops in some cases.

Avocado crops in the Canary Islands reached 2472 hectares in 2023, a 25.8% increase compared with 2020 [2]. The establishment of new plantations in this region requires the use of rootstocks from authorized nurseries to ensure plant health and certain agronomic properties. Currently, most plant nurseries produce rootstocks from seeds collected in fields of avocado mother-plants, which are classified at the race level based primarily on their morphological characteristics. However, these phenotypes are often difficult to distinguish, even for experienced specialists, particularly when dealing with hybrid individuals. This situation leads to orchards with low uniformity and generally low yields, averaging about 12 t/ha [3]. In this context, it is known that the genetic origin of the mother plant influences the resulting rootstocks’ tolerance to specific diseases, climatic conditions, and soil characteristics like salinity. Some of these traits are typically associated with one of the three main avocado horticultural races: *West-Indian*, *Guatemalan*, or *Mexican* [4,5].

The Canary Islands have agricultural soils with a high degree of degradation due to salinization, affecting about 57% of the cultivated land [6], mainly caused by irrigation with saline water extracted from volcanic groundwater, but also by sea spray or fertilizer accumulation. Comparable conditions occur in Mediterranean agricultural regions, which are also affected by salinity and water scarcity, with these areas being where most of the European avocado production is concentrated [7]. Unfortunately, avocado is one of the most salt-sensitive crops [8], and it is noteworthy that the physiological response of ‘Hass’ avocado to salinity is influenced by the rootstock [8,9,10]. Since the *West-Indian* rootstocks are more salt tolerant than *Guatemalan* or *Mexican* ones [11,12,13] and are also more resistant to the *Phytophthora cinnamomi* (Rands, 1922) pathogen [14], the regional government of the Canary Islands requires the use of rootstocks from this horticultural race to be explored [15]. Therefore, the development of molecular tools that allow for easy identification of *West-Indian* rootstocks, and their possible hybridization with *Guatemalan* and/or *Mexican* cultivars (i.e., *non-West-Indian* plants), is of great interest.

Single Sequence Repeats [16] and Single Nucleotide Polymorphisms [17] have been exploited to generate avocado race-specific markers, but with varying success rates and sometimes with controversial results [18]. Alternatively, the inter-Primer Binding Site (iPBS) strategy, which takes advantage of the conserved Primer Binding Site (PBS) sequence within retrotransposons with Long Terminal Repeat (LTR) to generate DNA fingerprints [19], has provided a good delimitation of avocado horticultural races, especially when focused on *West-Indian* cultivars [20]. Moreover, one advantage of iPBS is its ability to identify potential LTR sequences, which could then be used in combination with locus-specific primers to design Retrotransposon-Based Insertional Polymorphism (RBIP) molecular markers. These RBIP markers are a well-established tool [21], which can be used for the characterization of avocado horticultural races through low-cost, high-throughput standard PCR amplification.

In the present work, iPBS markers and LTRs previously characterized in *P. americana* [20] were used to design a set of RBIP primers. These tools were validated in a multiplex PCR assay and tested with a set of 58 avocado cultivars. The primers allow for the inexpensive and fast identification of avocado *West-Indian* cultivars and the detection of their hybrids.

## 2. Results

### 2.1. Suitability of RBIP Molecular Markers for Avocado Horticulture Race Identification

To develop new RBIP markers for *P. americana* horticultural race detection, eight different primer pairs were designed. This was performed using available LTRs along with the corresponding locus-specific sequences, which were previously described through iPBS analysis (Table 1).

The eight PaRBIP assays were assessed independently using genomic DNA samples from a selection of 15 avocado cultivars, including three representatives of the *Mexican* (*M*), *Guatemalan* (*G*), and *West-Indian* (*W*) pure races, as well as four *Guatemalan x Mexican* (*GxM*), and two *Guatemalan x West-Indian* (*GxW*) hybrids. Overall, the eight markers yielded discrete bands with the expected amplicon lengths (Table 1), showing 100% of specificity for the different avocado races, judging by the results obtained with the cultivars used as references (Figure 1), although with different levels of sensitivity.

PaRBIP-1 emerged as a potential *G*-specific marker, since the expected amplicon (227 bp) was observed in nearly all cultivars with a *G* genetic background, but not in the pure *M* or *W* cultivars (Figure 1). In contrast, PaRBIP-2 and PaRBIP-8 seem to be *M*-specific markers. On one hand, the expected PCR product for the PaRBIP-2 marker (283 bp) was detected in one pure *M* cultivar (Topa–topa) and in all *GxM* hybrids (Figure 1). On the other hand, the expected band for the PaRBIP-8 marker (510 pb) was found in all cultivars with a *M* genomic component. PaRBIP-1, PaRBIP-2, and PaRBIP-8 were treated as *non-West-Indian* (*non-W*) markers in this work. Surprisingly, PaRBIP-8 primers also produced an extra band (260 bp) that was amplified in all samples (Figure 1). Therefore, PaRBIP-8 was also used as a positive control to verify the suitability of the DNA for PCR amplification. Finally, results for markers PaRBIP-3, -5, -6, -C1, and -C2 indicate they are all specific to the *W* horticultural race. These five markers failed to amplify in pure *M* and *G* cultivars, and in *GxM* hybrids, but were frequently detected in pure *W* cultivars and *GxW* hybrids (Figure 1).

Considering the compatibility between the eight primer pairs and the ability to resolve the different amplicons by electrophoresis in agarose gels, seven RBIP markers were grouped into three multiplex PCRs, for which adjustments in the concentration of certain primer pairs were necessary (Table 1). Markers PaRBIP-1, -2 and -3 were assayed together, as were PaRBIP-5 with -C2, and PaRBIP-6 with -C1, whereas PaRBIP-8 remained as a singleplex PCR assay (Table 1). The results obtained with the three multiplex assays and the same set of 15 selected cultivars were consistent with previous findings (Figure 2).

### 2.2. Sensitivity of PaRBIP Markers in Detecting the West-Indian Genomic Component

The analysis of PaRBIP markers was expanded to a total of 58 avocado cultivars (Table 2) to test the sensitivity of the RBIP strategy in detecting detect both the *W* and the *non-W* (i.e., *G* or *M*) genomic components. A clear horticultural race assignation was found in the consulted literature for 37 of these cultivars. However, for the remaining cultivars, contradictory data was encountered (nine cases) or was not available (12 cases) (Table 2). Among the 46 cultivars with available race information, 38 showed a *non-W* genomic component (*G*, *M*, *GxM*, or *GxW*). The *non-W* component was detected in all of cultivars using the PaRBIP-1, -2 and/or -8 markers. This resulted in a 100% sensitivity level in detecting the *non-W* genomic component when this three-marker combination was used. Conversely, according to the literature, ten cultivars had a *W* genomic background (pure *W* cultivars or *GxW* hybrids). All these were successfully detected using the set of PaRBIP-3, -5, -6, -C1, and -C2 markers, achieving a 100% sensitivity for the detection of the *W* component when this five-marker combination was implemented (Table 2).

Furthermore, the results showed no detection of *non-W* component in Maoz, SS3, and VGR20/32/38 cultivars, which aligns with their previous classification as pure *W* cultivars [37,38]. However, cultivars Taro H25 and Taro H27, which were traditionally considered pure *W* cultivars [38], were found to have a *non-W* genomic component (Table 2). For Choquette and Fuchs-20, which are commonly identified as *GxW* hybrids [23,25,27,30,32], both the *W* and *non-W* components were detected by the corresponding PaRBIP marker set (Table 2). The same results were obtained for the Lula cultivar, which has been inconsistently classified as a *GxM* [27] and *GxW* hybrid [25,36].

Interestingly, the RBIP strategy revealed novel information about G.A.-13 and Lonjas, both previously classified as *M* cultivars [33,35]. While the existence of a *non-W* genomic component was confirmed, a *W* component was detected in both cultivars for the first time. This finding is consistent with the observation that G.A.-13 exhibits a high tolerance to salinity, similar to that of *W* rootstocks [33].

Finally, this study tested 11 cultivars for which no horticultural race information was found in the literature, but are classified as *W* cultivars by the agronomist who supplied us with the samples. Among these, A3, BA2, De La Verruga, Gallo 4, M1, T23, and V1 showed only the *W* genomic component. The remaining four cultivars—Gallo 2, Gallo 3, Julian, and Taro—showed both *W* and *non-W* components, revealing their hybridization with *G* and/or *M* cultivars.

## 3. Discussion

In this study, a set of eight novel RBIP markers was developed, using the iPBS markers previously described in *P. americana* [20] as a starting point. These tools represent a significant step forward for avocado rootstock production by facilitating the transition from phenotype-based identification to a more precise approach based on molecular markers. The main objective of our work was to identify pure *W* cultivars and their hybrids. However, it should be noted that the number of markers is not high enough to completely rule out all the *G* and/or *M* genomic components, but they still represent a powerful tool for the fast detection of *W* cultivars. The accurate identification of *W* rootstocks is critical for the long-term sustainability of avocado cultivation, particularly in regions with high soil salinity, such as the Canary Islands and Mediterranean regions. By enabling the accurate identification of *W* rootstocks, known for their chloride ion-excluding properties, our PaRBIP marker set provides a direct tool for growers to proactively select varieties genetically predisposed to tolerate high salt levels. Similarly, these markers could contribute to controlling the devastating root rot caused by *Phytophthora cinnamomi*. This selection of resilient varieties leads to healthier avocado trees, sustained yields, predictable outcomes in the field, and significant economic savings by helping to prevent losses caused by salt stress or this fungal pathogen.

Our research has shown that traditional classification of avocado rootstocks is often inaccurate due to widespread and unrecorded hybridization. Many cultivars previously considered “pure” are, in fact, hybrids, such as the cultivar G.A.-13. As stated by our results, the G.A.-13 is a complex hybrid with the three racial components. Moreover, the fact that G.A.-13, known for its salinity tolerance, has a *W* genetic component provides an explanation for this valuable trait. Our PaRBIP markers also have clarified the genetic makeup of some cultivars. For example, the cultivar Lula has been contradictorily classified as *GxM* [27] and *GxW* [25,36], but the PaRBIP markers have revealed that this cultivar possesses both *M* and *W* genomic components. The analysis of cultivars with complex pedigrees highlighted the ability of the PaRBIP markers to detect genetic introgression. For instance, Taro H25 and H27, which have been traditionally managed as pure *W* cultivars, showed a *non-West-Indian* genomic component, which indicates that these cultivars are hybrids. In these cases, a *W*-positive result confirms the presence of the marker-related *W* genome component but does not quantify the proportion of the *W* genome which is present in a hybrid cultivar. In complex multi-generational hybrids, a cultivar might retain a specific marker (i.e., *W*-positive), while losing other traits that confer the racial phenotypes (i.e., the salinity tolerance of the *W* race). Therefore, while these markers are robust for excluding material containing a *non-West-Indian* genomic component, the positive result for *W*-specific markers in complex hybrids should be interpreted with caution, since it could be the result of genetic introgression.

It should be noted that the fact that the PaRBIP-1 marker is not detected in the cultivar Lula does not mean that it lacks a *G* genetic component. A higher number of race-specific genetic markers are required to rule out the existence of a significative race-specific genetic component in the analyzed individual. In this sense, the PaRBIP markers presented in this work are dominant because they only detect the allelic variant that presents an LTR of a retrotransposon next to a specific genomic locus [21]. Therefore, despite the fact that the method does not allow for the quantification of the heterozygosity levels within hybrid individuals, the five *W*-specific markers used together are highly effective for certifying the presence of the *West-Indian* genomic component. Considering that the focus of this study is the development of a method for a fast and cost-effective screening of *West-Indian* rootstocks in commercial nurseries, we consider that it provides enough resolution to support downstream breeding decisions, especially when combined with traditional morphological screening. Our research group is currently working to identify the alternative allele (i.e., without the inserted retrotransposon) of PaRBIP markers and develop codominant ones, with the hope that these new alleles will be specific to the alternative race(s). In fact, this task has become much easier after the publication of the first haplotype-resolved genome of Hass [39]. This would increase the ability to detect race-specific genomic components.

The validation of the PaRBIP markers was conducted using germplasm collections maintained in Spain (the Canary Islands and Málaga). Therefore, for extending the applicability of our assays, it could be necessary to perform a validation with samples maintained in germplasms from Central and South America, due to the high genetic diversity found in these regions [40]. Additionally, regarding functional traits, our markers identify the presence of the *West-Indian* genomic component, and this race is widely documented to possess superior salinity tolerance and resistance to *P. cinnamomi*. However, it cannot be stated that PaRBIP markers are linked to loci involved in salinity tolerance or resistance to this pathogen.

Previous studies have developed race-specific codominant markers for avocado cultivars. Gross-German and Viruel [16] have reported 15 polymorphic simple sequence repeat (SSR) loci, with one or two alleles per marker being race-specific (seven *GxM*, two *M,* and six *W* markers). In addition, Ge et al. [17] have described eight single nucleotide polymorphisms (SNPs), with one variant in each locus being race-specific (four *G*, three *M,* and one *W* markers), and they have proposed the genotyping of these loci using real-time PCR and fluorescently labeled oligonucleotides. The advantages of the described procedure for the analysis of our set of PaRBIP markers include the fact that it does not require specialized instrumentation, such as a real-time PCR platform or a system for capillary electrophoresis, and avoids the use of expensive fluorescent labels. Furthermore, our PaRBIP markers have been validated for multiplex PCR, making genetic screening highly cost-effective, scalable, and practical for nurseries managing large volumes of seedlings. The initial cost of this testing is a minor investment compared to the multi-year losses that can result from a poorly performing orchard.

## 4. Materials and Methods

### 4.1. P. Americana Cultivars and Genomic DNA Purification

Leaf samples of 58 *Persea americana* cultivars were retrieved from Instituto Canario de Investigaciones Agrarias (ICIA. La Laguna, Canary Islands, Spain), Instituto de Hortofruticultura Subtropical y Mediterránea La Mayora (Málaga, Andalucía, Spain), and Agro-Rincón S.L (Los Realejos, Canary Islands, Spain) (Table 2).

Young leaves were collected from adult trees, without symptoms of disease, chlorosis, or wounds, and genomic DNA was purified as explained elsewhere [20]. Each DNA sample was diluted to a final concentration of 10 ng/µL in 10 mM Tris-HCl pH 8.0, and stored at −20 °C. From these stocks, working dilutions at 0.4 ng/µL were prepared in the same buffer, for PCR amplification.

### 4.2. Design of Oligonucleotides for PaRBIP Markers and PCR Amplification

From the LTR sequences of *P. americana* and associated locus-specific sequences identified by González-Carracedo et al. [20], eight primer pairs were designed for PCR amplification of each locus (Table 1), including one LTR-binding primer and one locus-specific primer. Evaluation of secondary structure and Tm calculations were carried out with the GeneRunner v6.5.52 software [22].

Singleplex PCRs were performed in a final volume of 20 μL containing 2 ng of genomic DNA template (H_2_O for negative controls), 1X Key Buffer (VWR, Radnor, PA, USA), 0.5 µg/µL BSA, extra MgCl_2_ (0.5 mM), 0.2 mM of each dNTP, 0.2 µM of each PaRBIP primer, and 0.5 U of Taq DNA polymerase (VWR). Reactions were incubated in a ProFlex PCR System (ThermoFisher, Waltham, MA, USA) with the following thermal profile: one initial denaturation step at 95 °C for 2 min; 35 amplification cycles with denaturation at 95 °C for 10 s, primer annealing at 60 °C for 10 s, and primer extension at 72 °C for 20 s (40 s in the case of PaRBIP-8 marker), with a final extension step carried out at 72 °C for 1 min. For multiplex PCR, the concentration of certain primers pairs was adjusted (Table 1).

Five microliters of each PCR were analyzed by electrophoresis in agarose gels (2%) prepared in 1X TBE buffer. Gels were run at 5 V/cm for 90 min and then stained in a 3X GelRed solution (Biotium, Fremont, CA, USA) for 30 min. Results were visualized under UV light using a ChemiDoc XR+ System (BioRad, Hercules, CA, USA).

## 5. Patents

In this paper, we refer to the methodology described in the Spanish patent application ES2957491A1, which details a diagnostic kit and method for identifying *Persea americana* horticultural races.

## Figures and Tables

**Figure 1 ijms-26-11510-f001:**
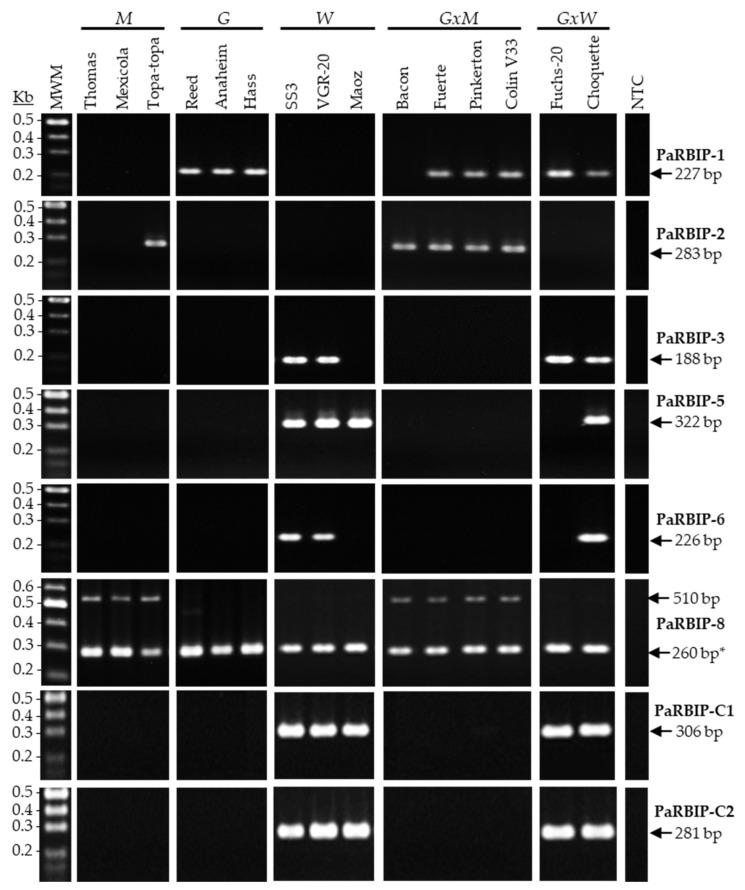
Singleplex RBIP analysis of *P. americana* cultivars, representatives of the three pure horticultural races and some hybrids. A set of 15 different *P. americana* cultivars were analyzed, with the PaRBIP primer pairs indicated at the right side. *M* (*Mexican*); *G* (*Guatemalan*); *W* (*West-Indian*); *GxM* (*Guatemalan x Mexican* hybrid); *GxW* (*Guatemalan x West-Indian* hybrid); Kb (Kilobases); MWM (Molecular weight marker); NTC (Non-template control); * Positive control for PCR.

**Figure 2 ijms-26-11510-f002:**
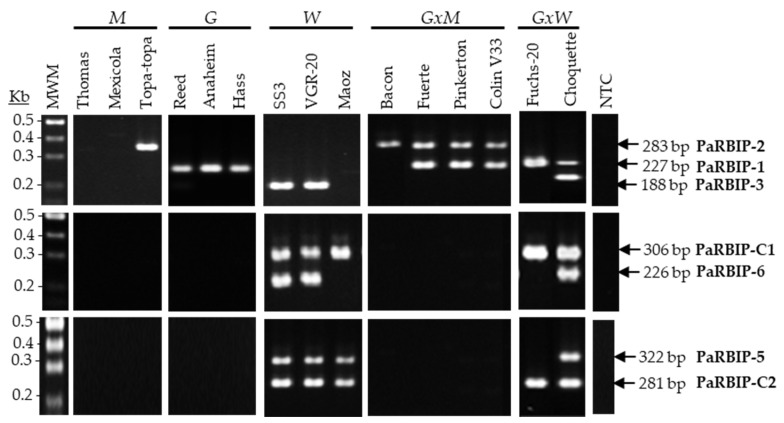
Multiplex RBIP analysis of *P. americana* cultivars, representatives of the three different pure horticultural races and some hybrids. A set of 15 different *P. americana* cultivars were analyzed, with the combination of PaRBIP primer pairs indicated at the right side of the figure. *M* (*Mexican*); *G* (*Guatemalan*); *W* (*West-Indian*); *GxM* (*Guatemalan x Mexican* hybrid); *GxW* (*Guatemalan x West-Indian* hybrid); Kb (Kilobases); MWM (Molecular weight marker); NTC (Non-template control).

**Table 1 ijms-26-11510-t001:** Primers designed for race-specific RBIP analysis in *P. americana*.

RBIP Assay	Primer ID	Sequence (5′-3′)	Tm (°C) ^1^	Expected Length (bp)	Multiplex Assay	Primer Conc. in Multiplex (µM)
PaRBIP-1	PaRBIP-1F	CCAACCAATCTATTTATTATGGAATCT	60.6	227	A	0.4
	PaRBIP-1R	CCAAGCTCTAAGAAAGGAAAACC	61.7			
PaRBIP-2	PaRBIP-2F	TGTCCCTCGTGGTTTATTCTATC	61.7	283	A	0.2
	PaRBIP-2R	AATGTAGGCTCTAAGAAAGGAAATAC	60.1			
PaRBIP-3	PaRBIP-3F	AGCTAACCTTGGAGCCTTCTC	62.5	188	A	0.2
	PaRBIP-3R	CTAGCTGGACTGGATTGATGG	62.0			
PaRBIP-5	PaRBIP-5F	TGTCGGGGTGACAAGATATTTC	62.8	322	B	0.3
	PaRBIP-5R	AACTCACCTATAAGGGTCTAATCAAC	60.9			
PaRBIP-6	PaRBIP-6F	CTATCCACTTCTTTGCGGACTAC	62.0	226	C	0.2
	PaRBIP-6R	CTCTATAGTCGATGTGGGACTCC	62.1			
PaRBIP-8	PaRBIP-8F	AGAAGATGGACAGTTCGGATCA	65.3	260 + 510	NA	0.2
	PaRBIP-8R	AACGAGAGTGGACGTTGACCT	65.1			
PaRBIP-C1	PaRBIP-C1F	TGCCCCTACATTTGGAGATTC	62.8	306	C	0.2
	PaRBIP-C1R	GATGGGTCATGGATGGCTAAC	63.2			
PaRBIP-C2	PaRBIP-C2F	ACGAGATTGGATAGCACCATGT	63.4	281	B	0.2
	PaRBIP-C2R	CCTTGAGGATTCACCATCATGT	62.8			

^1^ Melting temperature, calculated with Gene Runner v6.5.52 software [22]. NA: Not available.

**Table 2 ijms-26-11510-t002:** Results of the analysis of 58 avocado cultivars with PaRBIP markers.

Cultivar	Source	PaRBIP Markers								This Study	Previously Reported ^1^	References
		1	2	3	5	6	C1	C2	8			
A3	ICIA ^2^	0	0	1	0	1	1	1	0	*W*	N/A	N/A
Adi	La Mayora	1	0	0	0	0	0	0	0	*non-W* (*G*)	*G*/*GxM*	[23,24]
Anaheim	La Mayora	1	0	0	0	0	0	0	0	*non-W* (*G*)	*G*	[23,25,26,27,28]
Arona	Agro-Rincón	1	0	0	0	0	0	0	0	*non-W* (*G*)	*M*	[23,27]
BA2	ICIA	0	0	1	1	1	0	0	0	*W*	N/A	N/A
Bacon	Agro-Rincón	0	1	0	0	0	0	0	1	*non-W* (*M*)	*M*/*GxM*	[23,25,27,28,29]
BL-5552	La Mayora	1	1	0	0	0	0	0	1	*non-W* (*GxM*)	*GxM*	[29]
BL667	ICIA	1	0	0	0	0	0	0	1	*non-W* (*GxM*)	*GxM*	[29]
Choquette	ICIA	1	0	1	1	1	1	1	0	*non-W (G)xW*	*GxW*	[23,25,27,30]
Colin V-33	La Mayora	1	1	0	0	0	0	0	1	*non-W* (*GxM*)	*GxM*	[23,27]
Duke Parent	La Mayora	0	1	0	0	0	0	0	1	*non-W* (*M*)	*M*	[23,25,29]
Duke-7	La Mayora	1	1	0	0	0	0	0	1	*non-W* (*GxM*)	*M*	[14,29]
Eden	La Mayora	1	0	0	0	0	0	0	1	*non-W* (*GxM*)	*GxM*	[31]
Ettinger	La Mayora	0	1	0	0	0	0	0	1	*non-W* (*M*)	*GxM*	[23,25,27,29,30]
Fuchs-20	La Mayora	1	0	1	0	0	1	1	0	*non-W (G)xW*	*GxW*	[27,32]
Fuerte	Agro-Rincón	1	1	0	0	0	0	0	1	*non-W* (*GxM*)	*GxM*	[23,28,29]
G-6	La Mayora	0	1	0	0	0	0	0	1	*non-W* (*M*)	*M*	[29]
G.A-13	La Mayora	1	1	0	0	0	1	1	1	*non-W* (*GxM*)*xW*	*M*	[33]
Gallo 2	ICIA	1	0	0	1	0	0	0	0	*non-W (G)xW*	N/A	N/A
Gallo 3	ICIA	1	0	0	1	1	0	0	1	*non-W (GxM)xW*	N/A	N/A
Gallo 4	ICIA	0	0	1	0	1	1	1	0	*W*	N/A	N/A
Hass	ICIA	1	0	0	0	0	0	0	0	*non-W* (*G*)	*G*/*GxM*	[23,25,27,28,29]
Horshim	La Mayora	1	0	0	0	0	0	0	1	*non-W* (*GxM*)	*GxM*	[23,25]
Iriet	La Mayora	1	1	0	0	0	0	0	1	*non-W* (*GxM*)	*GxM*	[25,34]
Jim	La Mayora	0	1	0	0	0	0	0	0	*non-W* (*M*)	*M*/*GxM*	[23,25,27]
Julian	ICIA	0	1	1	1	1	1	1	0	*non-W (M)xW*	N/A	N/A
Lamb-Hass	ICIA	1	0	0	0	0	0	0	1	*non-W* (*GxM*)	*GxM*	[25,29]
Lonjas	La Mayora	1	0	1	1	0	0	0	0	*non-W (G)xW*	*M*	[35]
Lula	La Mayora	0	0	0	1	0	1	1	1	*non-W (M)xW*	*GxM*/*GxW*	[25,27,36]
M1	ICIA	0	0	0	1	0	1	0	0	*W*	N/A	N/A
Maoz	La Mayora	0	0	0	1	0	1	1	0	*W*	*W*	[14,23,27,37]
Mexicola	La Mayora	0	0	0	0	0	0	0	1	*non-W* (*M*)	*M*	[23,25,27,30]
Negra de la Cruz	La Mayora	1	1	0	0	0	0	0	1	*non-W* (*GxM*)	*M*/hybrid	[35,36]
OA-184	La Mayora	1	0	0	0	0	0	0	0	*non-W* (*G*)	*GxM*	[29]
Orotava	ICIA	1	1	0	0	0	0	0	1	*non-W* (*GxM*)	*G*	[23,27]
Pinkerton	Agro-Rincón	1	1	0	0	0	0	0	1	*non-W* (*GxM*)	*GxM*	[23,25,27,28,29,30]
Puebla	Agro-Rincón	1	0	0	0	0	0	0	1	*non-W* (*GxM*)	*M*/*GxM*	[23,25,27]
Reed	Agro-Rincón	1	0	0	0	0	0	0	0	*non-W* (*G*)	*G*	[23,25,26,27,28]
Rincoatl	La Mayora	0	1	0	0	0	0	0	1	*non-W* (*M*)	*M*	[35]
Rincon	ICIA	0	1	0	0	0	0	0	1	*non-W* (*M*)	N/A	N/A
Schmidt	La Mayora	0	1	0	0	0	0	0	1	*non-W* (*M*)	*G*/*M*	[26,27]
Scott	La Mayora	0	1	0	0	0	0	0	1	*non-W* (*M*)	*M*	[27]
Shepard	La Mayora	0	1	0	0	0	0	0	0	*non-W* (*M*)	*G*	[27]
SS3	Agro-Rincón	0	0	1	1	1	1	1	0	*W*	*W*	[38]
T23	ICIA	0	0	1	0	1	1	1	0	*W*	N/A	N/A
Taro	ICIA	1	0	1	0	1	0	0	0	*non-W (G)xW*	N/A	N/A
Taro H25	ICIA	1	0	1	0	1	0	0	0	*non-W (G)xW*	*W*	[38]
Taro H27	ICIA	1	0	1	0	1	0	0	0	*non-W (G)xW*	*W*	[38]
Thomas	La Mayora	0	0	0	0	0	0	0	1	*non-W* (*M*)	*M*	[27,29]
Topa–Topa	La Mayora	0	1	0	0	0	0	0	1	*non-W* (*M*)	*M*	[23,25,27,28,29]
Toro Canyon	La Mayora	0	1	0	0	0	0	0	1	*non-W* (*M*)	*M*	[29,36]
V1	ICIA	0	0	1	1	0	0	0	0	*W*	N/A	N/A
De La Verruga	ICIA	0	0	1	0	1	1	1	0	*W*	N/A	N/A
VGR20	ICIA	0	0	1	1	1	1	1	0	*W*	*W*	[38]
VGR32	ICIA	0	0	1	1	1	1	1	0	*W*	*W*	[38]
VGR38	ICIA	0	0	1	1	1	1	1	0	*W*	*W*	[38]
Waterhole	La Mayora	0	1	0	0	0	0	0	1	*non-W* (*M*)	*M*	[27,29]
Zutano	La Mayora	0	0	0	0	0	0	0	1	*non-W* (*M*)	*M*/*GxM*	[25,28,29]

^1^ N/A (not available); *M* (*Mexican*); *G* (*Guatemalan*); *W* (*West-Indian*); *GxM* (*Guatemalan x Mexican* hybrid); *GxW* (*Guatemalan x West-Indian* hybrid). ^2^ ICIA (*Instituto Canario de Investigaciones Agrarias*).

## Data Availability

The original contributions presented in this study are included in the article. Further inquiries can be directed to the corresponding authors.

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
