# Peer review of "Development of Molecular Tools to Identify the Avocado (Persea americana) West-Indian Horticultural Race and Its Hybrids"

_ijms, 2025, doi:10.3390/ijms262311510_

Round 1
Reviewer 1 Report
Comments and Suggestions for Authors
This study has developed eight retrotransposon-based insertion polymorphism (RBIP) molecular markers, which can efficiently identify West-Indian avocado cultivars and their hybrids. It addresses the key need for genetic identification of avocado rootstocks in salt-stressed regions. The experimental design is reasonable with sufficient data support, making the study worthy of publication. However, supplementary explanations are required for the following issues to enhance the integrity and rigor of the research:
- In the Abstract section, the text should not be divided into paragraphs; instead, it should be presented as a complete single paragraph, as shown in lines 29-30.
- The Results section is intended to describe the findings of the study and should not include references. Relevant citations should be moved to the Materials and Methods section or the Discussion section; please revise lines 94-95 accordingly.
- Figure 1 should be placed after line 105.
- Table 2 should be moved after line 153.
- For "ICIA" in the "Source" column of Table 2, since it appears for the first time, its full name should be used.
- In the References section, the proportion of literature published in the past three years should be increased.
Author Response
Dear Reviewer,
Please see the attachment.
Regards.

Reviewer 2 Report
Comments and Suggestions for Authors
This paper presents a useful method to identify avocado rootstock. The experiments presented can be carried out using standard lab equipment as the cost of materials is low. Therefore, there should be no major problem adapting this method widely. The limitation of the method is that the number of markers is not high enough to rule out race-specific components in the analyzed individuals. The authors mention this fact in their paper in the case of one cultivar, but perhaps they could state that this is perhaps true in other cases. Else I have only two suggestions concerning grammar and as I also work in the field of plant taxonomy would like to add the author of Phytophtora cinnamomi. I attached a copy of the pdf with my comments.

Author Response
Dear reviewer,
Please see the attachment.
Regards.

Reviewer 3 Report
Comments and Suggestions for Authors
The manuscript presents the development of eight novel Retrotransposon-Based Insertion Polymorphism (RBIP) markers for identifying West-Indian avocado (Persea americana) cultivars and their hybrids. The study addresses an important challenge in avocado breeding, particularly in regions such as the Canary Islands. The use of multiplex PCR combined with standard agarose gel electrophoresis makes the technique practical, cost-effective, and accessible without the need of specialized instrumentation. Having that said, there are several points where the manuscript could be improved.
First, the markers developed are dominant, detecting only the presence of a retrotransposon insertion, which limits the ability to identify heterozygous loci or quantify the contribution of multiple races in hybrids. A more detailed discussion of how this limitation may affect hybrid detection and breeding applications would strengthen the manuscript. Truly, the authors acknowledge this, but the manuscript could better discuss how this major limitation might affect the accuracy of hybrid detection and downstream breeding decisions.
Second, the cultivars analyzed are largely from Spain, which may limit the generalizability of the findings to avocado germplasm in Central or South America or other global growing regions. Without experimental validation, the link between marker presence and functional traits remains speculative. Thus, I suggest limiting this point to what can actually be said from the results.
Some cultivars exhibit complex hybrid backgrounds, and the manuscript could provide more discussion on how such complexity might affect marker interpretation.
The manuscript is also highly technical, with detailed PCR protocols, primer sequences, and multiple marker descriptions. Moving some details to Supplementary Materials would improve readability.
Finally, minor editorial issues, such as inconsistent references to “PaRIPB-8” instead of “PaRBIP-8,” should be corrected.
Author Response

(The authors gave the same response as above.)
